# Recovery of Filtered Particles by Elastic Flat-Sheet Membrane with Cross Flow

**DOI:** 10.3390/membranes11020071

**Published:** 2021-01-20

**Authors:** Manoka Miyoshi, Shintaro Morisada, Keisuke Ohto, Hidetaka Kawakita

**Affiliations:** Department of Applied Chemistry, Saga University, Saga 840-8502, Japan; goodfish1207@gmail.com (M.M.); morisada@cc.saga-u.ac.jp (S.M.); ohtok@cc.saga-u.ac.jp (K.O.)

**Keywords:** elastic membrane, filtration, recovery, cross flow, filtration residue

## Abstract

After filtration, filtered residue is recovered by a spoon, during which, the structure of the residue is destroyed, and the activity of the microorganism would be reduced. Thus, a more efficient recovery method of filtered residue is required. This study addressed the recovery method of filtered residue by the restoration of an elastic membrane, followed by cross flow. An elastic membrane composed of a copolymer of poly(ethylene glycol) diacrylate and polyacrylonitrile was prepared by photopolymerization. The pore diameter of the obtained membrane was about 10 μm. Silica particle (1 and 10 μm) and *Nannochloropsis* sp. (2 μm) suspension was filtered, demonstrating that silica particles of 10 μm were filtered perfectly, whereas the filtration percentage of 1 μm silica particles and *Nannochloropsis* sp. was lower. After the filtration, the applied pressure was released to restore the elastic membrane which moved the filtered particles up, then the filtered residue was recovered by cross flow above the membrane, demonstrating that 71% of the filtered 10 μm silica particles was recovered. The elastic behavior of the membrane, along with the cross flow, has the potential to be used as a technique for the recovery of the filtered residues. This proposed scheme would be used for the particle recovery of ceramics, cells, and microorganisms from a lab scale to a large-scale plant.

## 1. Introduction

During the filtration of cells and microorganisms with the filter, a cake layer of residue is formed on the pore [1,2]. To recover the filter residue on the membrane, it requires sweeping with a spoon. However, the structure and the activity of the filtered residues might be lost, and thus a novel technique for the recovery of the filtered residues is needed.

For the filtration and separation of particles dispersed in solution, so far, we have proposed the recovery method of particles being filtered by a spherical gel layer packed in a column. Colloidal suspension, as a separation target, was loaded on the spherical gel layer, and water was flowed through the column for the separation [3,4,5]. The larger colloidal particles were filtered at the top of the gel layer, and the smaller particles were transported to the inner domain of the column. To recover the large filtered colloidal particles, the applied pressure was released to restore the gel layer. The filtered residue was raised up by the elasticity of the gel layer. The raised particles were recovered by cross flow across the column [6]. However, the filtered particles in the inner part of the gel layer remained there.

Synthetic membrane and gel are composed of synthetic polymers. Photopolymerization is one of the methods used for polymerization. Photopolymerization can be applied to the local domain and to the surface. Ethylene glycol diacrylate is commonly used for many applications because ethylene glycol is hydrophilic, which prevents the hydrophobic interaction [7,8]. For instance, the membrane was modified with ethylene glycol for the separation of water and oil [9]. The membrane composed of polyethylene glycol was prepared in a microchannel to determine the deformation of the membrane by applied pressure of fluid flow [10]. The linear domain composed of polyethylene glycol has the characteristics of extension and shrinkage, applicable for the deformation of membrane. Direct observation of membrane and gel deformation in membrane modules is still difficult. Therefore, a mathematical model based on direct mechanical data of the membrane has been analyzed instead of the observation [11]. However, the effect of fluid flow in each pore was still not analyzed. So far, though the polymeric gel membrane in the microchannel was deformed by fluid flow in order to be observed directly [12,13], membrane deformation along with the particle filtration has not yet been analyzed and has not applied for actual separation.

In this study, elastic membrane as a flat sheet was prepared by photopolymerization. Colloidal suspensions were filtered by the elastic membrane, and the filtered residue was recovered by the restoration of the elastic membrane and cross flow, as shown in Figure 1. Concretely, (a) when colloidal suspension is filtered by the elastic membrane, the membrane was deformed by the applied pressure; (b) after releasing the applied pressure, the elastic membrane was restored to raise the filtered residues up; (c) raised residue was recovered by the cross flow above the membrane surface. In the case of cross flow, shear stress by fluid on the microfiltration and ultrafiltration membrane pore sweeps the filtered particles. Back flow is generated from the opposite direction to the filtered flow direction to wash out the residues. If the residue still remains on the membrane pore, a more effective method for the recovery is required. The originality of this study is that the membrane deformation with the elasticity and back flow, along with the cross flow, enhances the recovery efficiency of the filtered particles. Specifically, poly(ethylene glycol) diacrylate (Mn 575) and acrylonitrile was used as a monomer for the polymerization. Poly(ethylene glycol diacrylate) (PEGDA) has elasticity, because the oligomer has the intrinsic “spring-like” structure [14]. Added polyacrylamide can form the porous structure because of its high exclusion volume in the polymer matrix. Colloidal particles of silica particles (1 and 10 μm) and *Nannochloropsis* sp. (2 μm) were used. The rigid and soft particles of silica particles and *Nannochloropsis* sp., respectively, were filtered, and the recovery of the filtered residue by the elasticity and cross flow was examined.

## 2. Materials and Methods

### 2.1. Materials

Monomer of poly(ethylene glycol) diacrylate (PEGDA) (Mn = 575) and acrylonitrile was obtained from Sigma–Aldrich (St. Louis, MO, USA) and FUJIFILM Wako Pure Chemical Industries (Tokyo, Japan), respectively. Polyacrylamide (MW. 5000–6000 kDa), as a chemical for the pore formation, and 2-hydroxy-2-methylpropiophenone (HMP), as a photoinitiator, were purchased from FUJIFILM Wako Pure Chemical Corporation (Osaka, Japan) and Tokyo Chemical Industry Co. (Tokyo, Japan), respectively. The UV lamp used for the polymerization was Model UVGL-25 (Funakoshi Co., Ltd., Tokyo, Japan). Silica particles used had 1 and 10 μm diameter and were purchased from micromod (Partikeltechnologie GmbH, Rostok, Germany). *Nannochloropsis* sp. (Marine fresh, 0606) was obtained from Higashimaru Co., Ltd. (Kagoshima, Japan). An optical image of *Nannochloropsis* sp. is shown in Appendix A (Figure A1).

The cross flow for the permeation through the membrane was lab-made equipment. The filter holder (diameter: 5.9 cm), O ring, pressure gauge, and peristat pump were from Merck KGaA(SX0004700), MonotaRO Co., Ltd. (Amagasaki, Japan) (ARP568-30A, diameter 41 mm, nitrile rubber, wire diameter 1.78 mm), Nagano Keiki Co., Ltd. (Higashimagome, Japan), and Tokyo Rikakikai Co., Ltd. (Tokyo, Japan) (ROLLER PUMP, RP-2000), respectively. Other chemicals were of analytical grade or higher. UV–Vis for the determination of the concentration of colloidal particles was JASCO Corporation (V-630BIO, Tokyo, Japan). 

### 2.2. Preparation of Elastic Membrane

PEGDA, acrylonitrile, and HMP were measured at the mass shown in Table 1 and mixed with polyacrylamide (PAA) solution (10 g/L). After dissolution, nitrogen gas was purged to the solution to remove oxygen for 1 min. The solution was added to a petri dish and sealed at the nitrogen atmosphere. A UV lamp at the wavelength of 365 nm was irradiated to the solution for polymerization for one hour at room temperature in a dark room. The heat flux was set at 0.72 × 10^−3^ W/cm^2^. After the polymerization, the obtained membrane was washed with water repeatedly. The membrane without added PAA was also prepared with the same procedure for comparison. The porosity of the membrane was determined by the weight change before and after the membrane was immersed in water. To determine the elastic modulus of the membrane, the dynamic ultra micro hardness tester (DUH-211S, Shimadzu, Kyoto, Japan) was performed three times.

### 2.3. Permeation of Water and Colloidal Particle Suspension through Elastic Membrane

The permeation apparatus is illustrated in Figure 2. The cross flow for the permeation through the membrane was lab-made equipment, as shown in Figure A2 in Appendix A. The prepared flat-sheet membrane was cut at the diameter of 47 mm. The membrane was set at the filter folder and water was flowed at the prescribed pressure to calculate the water flux, m^3^/m^2^·min. The pressure was controlled with the lever of the peristat pump to set a constant pressure of 0.010 MPa. The effluent volume was measured for a prescribed time, and the mass was changed to the effluent volume by a density of 1000 kg/m^3^. After the water permeation, the pressure was again applied for the determination of water flux for repeated measurements.

Suspensions of silica particles (1 and 10 μm, 0.1 g/L) and *Nannochloropsis* sp. (2 μm, 0.1 g/L) were individually permeated through the membrane at 0.01 MPa. The concentration of colloidal particles was determined by the wavelength of 600 nm of UV–Vis (JASCO Corporation, V-630BIO, Japan). 

### 2.4. Recovery of Filtered Particles by Elastic Membrane

Suspensions of silica particles (10 μm, 0.1 g/L) and *Nannochloropsis* sp. (2 μm, 0.1 g/L) were firstly permeated to the elastic membrane at 0.01 MPa for 30 min by the apparatus of Figure 2. After filtration, the applied pressure was released to make the filtered residues rise up. Subsequently, cross flow at 2760 mL/h above the membrane surface was performed for 30 s. The concentration in each effluent and recovered beaker, at 8 and 9 in Figure 2, respectively, was determined by UV–Vis. The percentage of filtered particle was defined as follows.

Percentage of filtered particle [%] = 100 (1 − (amount of filtered particle))/(initial amount of particle).

## 3. Results and Discussion

### 3.1. Preparation of Elastic Membrane

Poly(ethylene glycol) diacrylate and acrylonitrile were polymerized with the photopolymerization by adding PAA. Here, the polymerization of poly(ethylene glycol diacrylate) and acrylonitrile was already confirmed by infrared in the previous published papers [15]. The SEM image of the obtained elastic flat-sheet membrane was shown in Figure 3. The obtained membrane had the diameter and thickness of 6.5 cm and 1.8 mm, respectively. The outer edge of the membrane had the thickness of 2.0 mm because the monomer transferred to the wall of the petri dish via flow convection generated by surface tension. The porosity determined was 73%. The surface of the membrane was observed, as shown in Figure 3. The three different domains of the membrane were observed. Acrylonitrile, PAA, and PEGDA have the role of membrane matrix, pore formation, and elasticity of the membrane, respectively. Ethylene glycol in PEGDA especially has the ability of extension and shrinkage because of the spring structure of ethylene glycol. The pore of the membrane was 1–10 μm, applicable for the filtration of appropriated particles.

The elastic modulus of the obtained membrane with PAA was 25.2 N/mm^2^. Drira and Yadavalli determined the mechanical strength of poly(ethylene glycol) hydrogel by atomic force microscopy, demonstrating that the gel had the elastic modulus of 1.3 N/mm^2^ [16]. As the elasticity was dependent on the polymerization conditions as well as used monomers, the obtained membrane had higher elasticity due to the presence of PAA and macro-pore formation.

### 3.2. Filtration of Each Particle through Elastic Membrane

Flux changes at different pressures through the elastic flat-sheet membrane prepared with and without adding PAA is shown in Figure 4. The experiments were performed three times to check the reproducibility. The membrane without PAA showed the linear increment of flux with increasing applied pressure, whereas the membrane with PAA had the non-linear increase, demonstrating that the flat-sheet membrane would exhibit the deformation of the membrane because of the pressure. Figure 4 shows that the membrane with PAA represented the higher flux, demonstrating that the macromolecules such as PAA have the higher exclusion volume to form the large pore in the membrane. After the washing, a part of PAA would be leaked from the membrane. The repeated permeation of water to the membrane was performed, as Figure 4 shows, demonstrating that the flux obtained has reproducible data. Thus, the membrane had a reversible deformation against the applied hydrodynamic pressure.

Silica suspension (1 and 10 μm) and *Nannochloropsis* sp. (2 μm) suspension were individually permeated through the elastic membrane with PAA to determine the permeation flux and concentration change in the effluent, as shown in Figure 5. In the case of silica suspension, the flux and concentration in the effluent decreased, whereas in the case of *Nannochloropsis* sp., they increased and levelled off. In the case of 1 μm of silica, as shown in Figure 5, the flux decreased to show the lower concentration in the effluent than the initial concentration. This is due to the filtration of silica and the transfer of silica in the membrane by convection to the effluent. Twenty percent of silica (1 μm) was filtered in the membrane, resulting in a reduction in the permeation flux. In the case of 10 μm of silica, the flux decreased and the concentration in the effluent was decreased because of the deposition of silica on the membrane.

The surface of the membrane after the filtration was observed by SEM, as shown in Figure 6. The image of 10 μm of silica showed the filtered particles on the membrane surface. An amount of 1 μm of silica would be filtered on the membrane, and a part of the silica filtered in the membrane as a depth filtration. Figure 6c shows that some of the filtered *Nannochloropsis* sp. was observed on the membrane.

### 3.3. Recovery of Filtered Particles by Restoring the Elastic Membrane and Cross Flow

The filtered silica (10 μm) and *Nannochloropsis* sp. (2 μm) by the elastic membrane was recovered by the elasticity and cross flow, as shown in the scheme of Figure 1. Elution percentage, percentage remaining in the membrane and recovered percentage by cross flow are summarized in Table 2. The filtration percentage was higher for silica (10 μm). The recovery percentage of silica (10 μm) by cross flow was 71%. Silica was filtered on the membrane due to its larger size, and a large amount of silica filtered was raised up by the elasticity of the membrane, followed by the recovery by the cross flow. In the case of *Nannochloropsis* sp., the recovery percentage by cross flow was 0.93%. *Nannochloropsis* sp. at the smaller size passed through the membrane, resulting in the lower recovery percentage. During the filtration, the particles filtered were captured via adhesion, showing that the interaction of the target and the membrane should be considered for high recovery percentage by raising the filtered particles up by the elasticity of the membrane and cross flow.

## 4. Conclusions

The recovery scheme of filtered particles by the elasticity of the flat-sheet membrane has been proposed. Acrylonitrile and poly(ethylene glycol) diacrylate were photopolymerized by adding polyacrylamide. Ethylene glycol had a role of spring for elasticity and polyacrylamide formed a pore of the flat-sheet membrane. The observation by SEM demonstrated that the membrane had a pore of 10 μm. For the recovery of filtered particles, the water was permeated from the bottom of the membrane to raise the filtered particles up, and cross flow above the membrane was applied for the recovery of the filtered particles. Silica at 10 μm had a high filtration percentage and could be recovered by the elasticity of the membrane and cross flow. At present, the particles filtered by the membrane are recovered by a spoon by hand. The proposed method has the potential to recover the filtered particles by using the porous membrane.

## Figures and Tables

**Figure 1 membranes-11-00071-f001:**
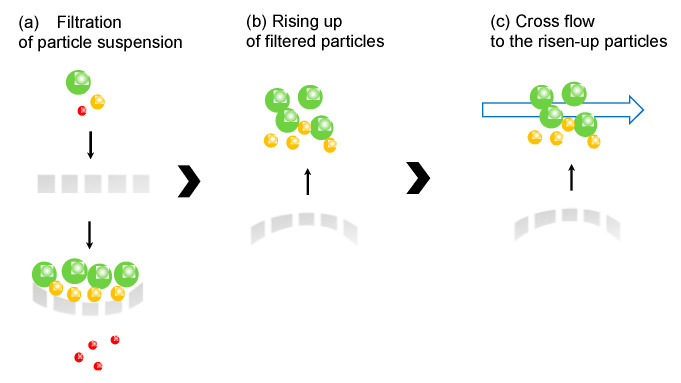
Illustrated image of the recovery of filtered particles by elasticity of the membrane and cross flow.

**Figure 2 membranes-11-00071-f002:**
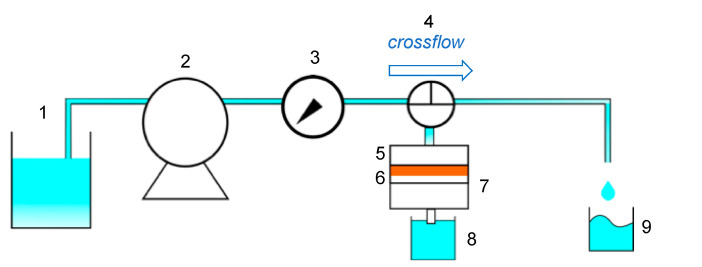
Cross flow apparatus of elastic membrane. 1: Feed solution, 2: pump, 3: pressure gauge, 4: three-way cock, 5: filter holder, 6: elastic membrane, 7: O ring, 8: elution, 9: recovery solution by cross flow.

**Figure 3 membranes-11-00071-f003:**
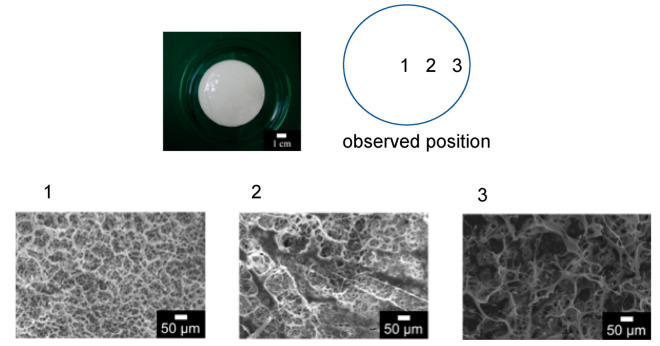
SEM images of the elastic membrane.

**Figure 4 membranes-11-00071-f004:**
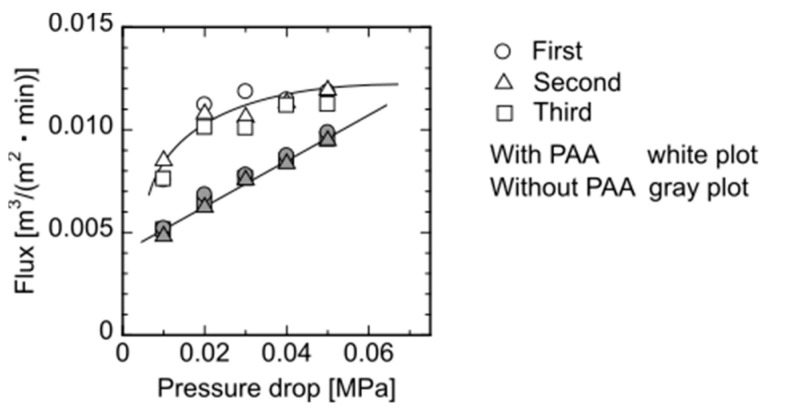
Steady-state flux of the membrane as a function of applied pressure. Each pressure was performed three times for reproducibility.

**Figure 5 membranes-11-00071-f005:**
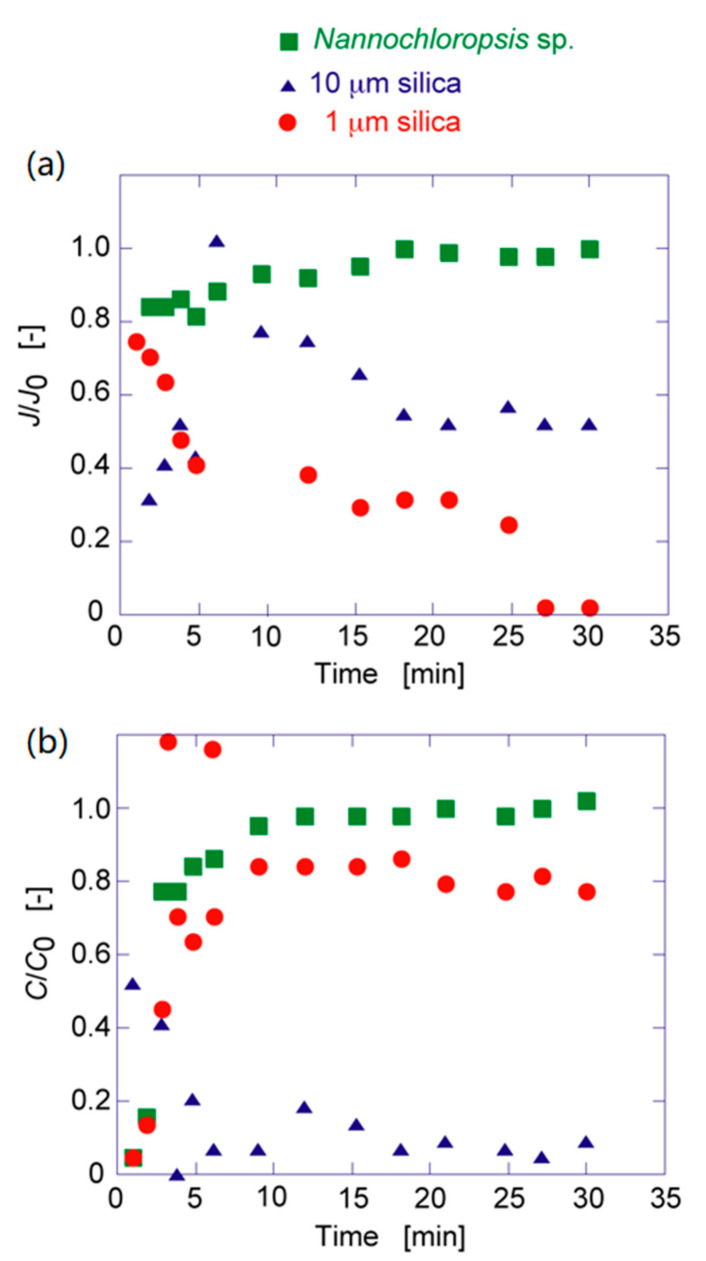
Time course curves of (**a**) flux ratio and (**b**) concentration ratio of each particle to the prepared elastic membrane.

**Figure 6 membranes-11-00071-f006:**
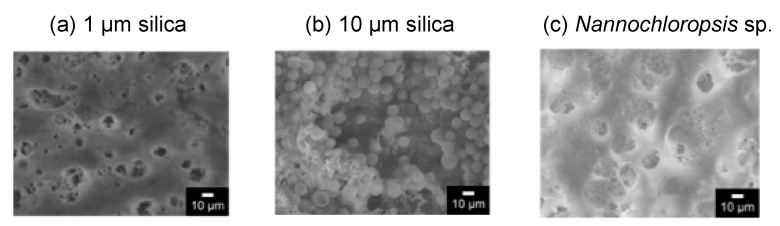
SEM images of the surface of the membrane after the filtration of each particle. (**a**) 1 μm silica; (**b**) 10 μm silica; (**c**) *Nannochloropsis* sp.

**Table 1 membranes-11-00071-t001:** Preparation of elastic membrane.

Chemicals	Amount of Chemicals	
	Mass [g]	Mole [mol]
Acrylonitrile	0.90	0.017
Poly(ethylene glycol) diacrylate (n = 10)	2.0	0.0034
Polyacrylamide	0.080	1.4 × 10^−8^
Distilled water	7.9	0.44
2-Hydroxy-2-methylpropiophenenone	0.050	0.00030

**Table 2 membranes-11-00071-t002:** Percentage of filtered particles using elasticity of the membrane and crossflow.

	Percentage of Particle [%]		
Filtered particles	elution	remaining in the membrane	recovered by crossflow
10 mm silica	14	15	71
2 mm *Nannochloropsis* sp.	93	6.1	0.93

## Data Availability

Data is contained within the article.

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
