# Peer review of "Recovery of Filtered Particles by Elastic Flat-Sheet Membrane with Cross Flow"

_membranes, 2021, doi:10.3390/membranes11020071_

Round 1

Reviewer 1 Report

In this paper, the authors report the results of a study on efficiently separating particulate matter through cross-flow in a filtration process using a porous elastic flat-sheet membrane. Although it is judged that the structure of this work is rather simple, it will be possible to publish after completing the corrections on the following points.

-(Introduction) The originality of this study should be clearly presented. A more detailed literature review for the previous researches is required.

-(P 2, line 49) "polymerized" -> "fabricated" or "prepared"

-(P2, lines 58-59) "Poly(ethylene glycol diacrylate) (PEGDA) has the ability..." Please notate a reference to this sentence. In addition, PEGDA is better to be notated as oligomer rather than monomer.

-(P2, line 69) Was 2-hydroxy-2-mthylpropiophenone used as a photoinitiator? If so, it should be stated that it was used as a photoinitiator.

-The mechanism of pore formation by adding polyacrylamide should be explained in the text.

-The reaction scheme of the photo-polymerization should be added in the manuscript and the FT-IR spectra are also needed to prove the synthesis.

-(P 3, line 126) How was the porosity of the prepared membrane measured? The pore size evaluation from the SEM images is very inaccurate. Please use an accurate method to measure and present the characteristics of the pores.

-(P 4, lines 136-137) It was discussed that the non-linear increase in the flux with increasing pressure was due to the deformation of the membrane. The authors should report additional experimental results indicating whether this deformation is reversible or not.

- Mechanical properties of the prepared elastic membrane should be supplied (modulus, tensile strength, etc.) in the revised manuscript.

-The graphs in Figure 5 are less readable and need to be corrected.

Author Response

Reviewer1

In this paper, the authors report the results of a study on efficiently separating particulate matter through cross-flow in a filtration process using a porous elastic flat-sheet membrane. Although it is judged that the structure of this work is rather simple, it will be possible to publish after completing the corrections on the following points.

[Query] (Introduction) The originality of this study should be clearly presented. A more detailed literature review for the previous researches is required.

[Answer] As the reviewer’s suggestion, the following sentences regarding the originality are inserted to the text,

In case of cross flow, shear stress by fluid on the microfiltration and ultrafiltration membrane pore sweep the filtered particle. Back flow was generated from the opposite direction to the filtered-flow direction to wash out the residues. If the residue still remained on the membrane pore, the more effective method for the recovery is required. The originality of this study is that the membrane deformation with the elasticity and back flow along with the cross flow enhances the recovery efficiency of the filtered particles.

[Query] (P 2, line 49) "polymerized" -> "fabricated" or "prepared"

[Answer] As the reviewer’s comments, ”polymerized” is revised to ”prepared”. Thank you very much.

[Query] (P2, lines 58-59) "Poly(ethylene glycol diacrylate) (PEGDA) has the ability..." Please notate a reference to this sentence. In addition, PEGDA is better to be notated as oligomer rather than monomer.

[Answer] As the reviewer’s comments, some references about "Poly(ethylene glycol diacrylate) (PEGDA).....” are inserted to the text, as follows,

Zhong C.; Wu, J.; Reinhart-King; Chu, C.C. Synthesis, characterization and cytotoxicity of photo-crosslinked maleic chitosan-polyethylene glycol diacrylate hybrid hydrogels, Acta Biomaterialia 2010, 6, 3908-3918.

As the reviewer said, ‘monomer’ is changed to ‘oligomer’.

[Query] (P2, line 69) Was 2-hydroxy-2-mthylpropiophenone used as a photoinitiator? If so, it should be stated that it was used as a photoinitiator.

[Answer] 2-hydroxy-2-mthylpropiophenone is a photoinitiator. As the reviewer’s suggestion, the following sentence is inserted to the text.

as a photoinitiator.

[Query] The mechanism of pore formation by adding polyacrylamide should be explained in the text.

[Answer] Figure 4 demonstrated that addition of PAA enhanced the flux of water. Figure 4 showed that the membrane with PAA represented the higher flux, demonstrating that the macromolecules such as PAA has the higher exclusion volume to form the large pore in the membrane. After the washing, a part of PAA would be leaked from the membrane. The following sentences are inserted to the text,

Figure 4 showed that the membrane with PAA represented the higher flux, demonstrating that the macromolecules such as PAA has the higher exclusion volume to form the large pore in the membrane. After the washing, a part of PAA would be leaked from the membrane.

[Query] The reaction scheme of the photo-polymerization should be added in the manuscript and the FT-IR spectra are also needed to prove the synthesis.

[Answer] As the reviewer’s suggestion, the following scheme of polymerization is mentioned to the Appendix in the text,

Figure A2. Preparation scheme of elastic membrane

The photo-polymerization of poly(ethylene glycol diacrylate) and acrylonitrile was confirmed by the previous published papers. The following sentences are inserted to the text,

Kang, G.; Cao Yiming.; Zhao, H.; Yuan, Q. Preparation and characterization of crosslinked poly(ethylene glycol) diacrylate membranes with excellent antifouling and solvent-resistance properties, J. Membrane Sci. 2008, 318, 227-232.

The polymerization of poly(ethylene glycol diacrylate) and acrylonitrile was already confirmed by infrared in the previous published papers.

[Query] (P 3, line 126) How was the porosity of the prepared membrane measured? The pore size evaluation from the SEM images is very inaccurate. Please use an accurate method to measure and present the characteristics of the pores.

[Answer] Porosity of the membrane was determined by the weight change before and after the membrane immersed in water. The determination of the pore by mercury intrusion was not performed, but the filtration of particles, silica, showed the pore of the membrane. The following sentences are inserted to the text,

Porosity of the membrane was determined by the weight change before and after the membrane immersed in water.

[Query] (P 4, lines 136-137) It was discussed that the non-linear increase in the flux with increasing pressure was due to the deformation of the membrane. The authors should report additional experimental results indicating whether this deformation is reversible or not.

[Answer] As the reviewer’s comments, the reversible deformation of the elastic membrane was mentioned as following sentence in the text,

The repeated permeation of water to the membrane was performed, as Figure 4, demonstrating that the flux obtained has the reproducible data. Thus, the membrane had the reversible deformation against the applied hydrodynamic pressure.

[Query] Mechanical properties of the prepared elastic membrane should be supplied (modulus, tensile strength, etc.) in the revised manuscript.

[Answer] As the reviewer’s comments, the mechanical elasticity was determined by mechanical equipment. The experimental condition and results are inserted to the text, as mentioned below,

Experimental: To determine the elastic modulus of the membrane, the dynamic ultra micro hardness tester (DUH-211S, Shimadzu, Japan) was performed to three times.

Results: The elastic modulus of the obtained membrane with PAA was 25.2 N/mm2. Drira and Yadavalli determined the mechanical strength of poly(ethylene glycol) hydrogel by atomic force microscopy, demonstrating that the gel had the elastic modulus of 1.3 N/mm2. As the elasticity was dependent on the polymerization conditions as well as used monomers, the obtained membrane had the higher elasticity due to the presence of PAA and macro pore formation.

Drira, Z.; Yadavalli, V.K. Nanomechanical measurements of polyethyeleglycol hydrogels using atomic force microscopy, J. Mechanic. Behavior Biomed. Mater. 2013, 18, 20-28.

[Query] The graphs in Figure 5 are less readable and need to be corrected.

[Answer] As the reviewer’s suggestion, Figure 5 was illustrated again. Thank you for your suggestion.

Figure 5. Time course curves of (a) flux ratio and (b) concentration ratio of each particle through the prepared elastic membrane.

Reviewer 2 Report

The authors reported on the application of a elastic flat-sheet filtration method. The manuscript is correctly written and English is generally good. However, the paper is not correctly structured in that some parts are not treated with enough care, as described below. Most importantly, conclusions are not supported the experimental results, which are limited to experiment method. Thus I suggest major revision. I have the following comments:

1. There is no mention about how to measure percentage of filtered particles and porosity of elastic membrane.
2. This manuscript has no consideration of elastic sheet deformation due to pressure. Elastic membrane should be considered the operation pressure.
3. Can you say that the experimental device is an experiment using cross flow?
4. Pressure drop was measured in section 3.2. How did you measure the pressure drop? I think you should mention the way of measurement.
5. Figure 5 should be upgraded.(Image quality and typo)

Author Response

Reviewer2

The authors reported on the application of an elastic flat-sheet filtration method. The manuscript is correctly written and English is generally good. However, the paper is not correctly structured in that some parts are not treated with enough care, as described below. Most importantly, conclusions are not supported the experimental results, which are limited to experiment method. Thus I suggest major revision. I have the following comments:

  1. [Query] There is no mention about how to measure percentage of filtered particles and porosity of elastic membrane.

[Answer] As the reviewer’s suggestion, percentage of filtered particle was inserted to the text, as below,

Percentage of filtered particle [%] = 100 (1- (amount of filtered particle)/(initial amount of particle)

  1. [Query] This manuscript has no consideration of elastic sheet deformation due to pressure. Elastic membrane should be considered the operation pressure.

[Answer] Thanks to the reviewer’s suggestion, the elastic modulus was determined by the mechanical equipment. The following sentences were inserted to the experimental, and results and discussion sections, as follows,

Experimental: To determine the elastic modulus of the membrane, the dynamic ultra micro hardness tester (DUH-211S, Shimadzu, Japan) was performed three times.

Results: The elastic modulus of the membrane was 25.2 N/mm2. Drira and Yadavalli determined the mechanical strength of poly(ethylene glycol) hydrogel by atomic force microscopy, demonstrating that the gel had the elastic modulus of 1.3 N/mm2. As the elasticity was dependent on the polymerization conditions as well as used monomers, the obtained membrane had the higher elasticity due to the presence of PAA and macro pore formation.

Drira, Z.; Yadavalli, V.K. Nanomechanical measurements of polyethyeleglycol hydrogels using atomic force microscopy, J. Mechanic. Behavior Biomed. Mater. 2013, 18, 20-28.

  1. [Query] Can you say that the experimental device is an experiment using cross flow?

[Answer] The cross flow to the membrane was lab-made equipment. The flow direction was changeable with the three-way cock. As the reviewer suggestion, the following sentence is inserted to the text,

The cross flow for the permeation through membrane was lab-made equipment.

  1. [Query] Pressure drop was measured in section 3.2. How did you measure the pressure drop? I think you should mention the way of measurement.

[Answer] As the reviewer said, the applied pressure, at 0.01 MPa, was controlled with the peristat pump to permeate colloidal suspensions. The following sentences are inserted to the text,

The pressure was controlled with the lever of peristat pump to set a constant pressure of 0.01 MPa.

  1. [Query] Figure 5 should be upgraded. (Image quality and typo)

[Answer] Thanks to the reviewer’s comments, Figure 5 was again illustrated.

Reviewer 3 Report

*) The abstract, as far as the work is concerned, is well written and sufficiently articulated. There is also a brief quantitative presentation of the results. However, there is a lack of potential future developments. This would help broaden the visibility of the paper.

*) The introduction seems to consist of parts that are disconnected from each other. It is advisable to introduce some linking phrases between the individual parts.

*) At the end of the introduction it is desirable that the description of the rest of the paper is inserted in order to introduce the reader in what will be the reading of the rest of the paper.

*) The text has some typos. Please reread the paper carefully in order to eliminate them.

*) Section 2.1 would seem to be a list of specific materials used in the work so that the reading of this section is rather sterile. Perhaps a few additional comments would enrich the text which, in itself, appears too poor.

*) Was the work performed through software, or is the device shown in Figure 2 present in the Laboratory? If so, Figure 2 should be replaced by a photograph showing the laboratory equipment.

*) The text mentions any problems of elastic deformation of the membrane. So, in support of this thesis, it would be better to insert a few sentences that underline this aspect by mentioning that scientific research is currently working hard in the development of physical-mathematical models that simulate the behavior of the membrane in the most varied operating conditions. In this regard, it is advisable to include the following works in the bibliography:

doi: 10.1016/B978-0-08-047488-5.00018-6

doi: 10.1515/caim-2017-0009

doi: 10.1271/journal.pone.0067708

Author Response

Reviewer3

[Query] *) The abstract, as far as the work is concerned, is well written and sufficiently articulated. There is also a brief quantitative presentation of the results. However, there is a lack of potential future developments. This would help broaden the visibility of the paper.

[Answer] Thanks to the reviewer’s comments, the following sentences are inserted to Abstract,

This proposal scheme would be used for the particle recovery of ceramics, cell, and microorganism from a lab scale to the large-scale plant.

[Query] *) The introduction seems to consist of parts that are disconnected from each other. It is advisable to introduce some linking phrases between the individual parts.

[Answer] Thanks to the reviewer’s comments, to connect the paragraph, the following sentences are inserted to Introduction,

First in 2nd paragraph: For the filtration and separation of particle dispersed in solution,

First section in 3rd paragraph: Synthetic membrane and gel was composed of synthetic polymer.

[Query] *) At the end of the introduction it is desirable that the description of the rest of the paper is inserted in order to introduce the reader in what will be the reading of the rest of the paper.

[Answer] Thanks to the reviewer’s comments, the following sentences are inserted to the introduction in the text,

In case of cross flow, shear stress by fluid on the microfiltration and ultrafiltration membrane pore sweeps the filtered particle. Back flow is generated from the opposite direction to the filtered-flow direction to wash out the residues. If the residue still remained on the membrane pore, the more effective method for the recovery is required. The originality of this study is that the membrane deformation with the elasticity and back flow along with the cross flow enhances the recovery efficiency of the filtered particles.

[Query] *) The text has some typos. Please reread the paper carefully in order to eliminate them.

[Answer] Thanks to the reviewer’s comments, some typo was revised.

[Query] *) Section 2.1 would seem to be a list of specific materials used in the work so that the reading of this section is rather sterile. Perhaps a few additional comments would enrich the text which, in itself, appears too poor.

[Answer] As the reviewer said, the blue character was inserted for easy understanding,

2.1. Materials

Monomers of poly(ethylene glycol) diacrylate(PEGDA)(Mn = 575) and acrylonitrile were obtained from Sigma-Aldrich and FUJIFILM Wako Pure Chemical Industries, respectively. Polyacrylamide (Mw. 5,000-6,000 kDa) , as a chemical for the pore formation, and 2-hydroxy-2-methylpropiophenone, as a photo-initiator, were purchased from FUJIFILM Wako Pure Chemical Corporation and Tokyo Chemical Industry Co., respectively. UV lamp used for the polymerization was Model UVGL-25 (Funakoshi Co., Ltd., Japan). Silica particle used had 1- and 10-mm diameter and was purchased from micromod (Partikeltechnologie GmbH, Germany). Nannochloropsis sp. (Marine fresh, 0606) was obtained from Higashimaru Co., Ltd., Japan. Optical image of Nannochloropsis sp. was shown in Appendix (Figure A1).

The cross flow for the permeation through membrane was lab-made equipment. Filter holder (diameter: 5.9 cm), O ring, pressure gauge, and peristat pump were from Merck KGaA(SX0004700), MonotaRO Co., Ltd. (ARP568-30A, diameter 41 mm, nitrile rubber, wire diameter 1.78 mm), Nagano Keiki Co., Ltd, and Tokyo Rikakikai Co., Ltd. (ROLLER PUMP, RP-2000), respectively. Other chemicals were of analytical grade or higher. UV-Vis for the determination of concentration of colloidal particle was JASCO Corporation (V-630BIO, Japan).

[Query] *) Was the work performed through software, or is the device shown in Figure 2 present in the Laboratory? If so, Figure 2 should be replaced by a photograph showing the laboratory equipment.

[Answer] The cross flow for the permeation through membrane was lab-made equipment. The equipment was imaged and shown in Figure A3. Thank you for your comments.

Figure A3. Picture of permeation apparatus

[Query] *) The text mentions any problems of elastic deformation of the membrane. So, in support of this thesis, it would be better to insert a few sentences that underline this aspect by mentioning that scientific research is currently working hard in the development of physical-mathematical models that simulate the behavior of the membrane in the most varied operating conditions. In this regard, it is advisable to include the following works in the bibliography:

[Answer] As the reviewer’s comments, the following sentences are inserted to the text,

Direct observation of membrane and gel deformation in membrane module is still difficult. Therefore, mathematical model based on direct mechanical data of the membrane has been analyzed instead of the observation. However, the effect of fluid flow in each pore was still not analyzed. So far, though polymeric gel membrane in microchannel was deformed by fluid flow to be observed directly, membrane deformation along with the particle filtration has not yet analyzed not applied for the actual separation.

Persson, K.M.; Gekas, V.; Trägårdh G. Study of membrane compaction and its influence on ultrafiltration water permeability, J. Membrane Sci. 1995, 100, 155-162.

Islam, M.A.; Stoicheva, R.N.; Dimov, A. An investigation on the deformational properties of porous poly(vinyl chloride) and co-poly(butadiene-acrylonitrile) blend membranes, J. Membrane Sci. 1996, 118, 9-15.

Cappello, J.; d’Herbemont, V.; Lindner, A.; du Roure, O. Microfluidic in-situ measurement of Poisson’s ratio of hydrogels, Micromachines2020, 11, 318.

Round 2

Reviewer 1 Report

The authors have carefully revised the manuscript according to the referees’ comments. In my opinion, this manuscript could be accepted for publication in Membranes.

Reviewer 2 Report

I think this paper deserves to be published in the journal.